# Successful SARS-CoV-2 mRNA Vaccination Program in Allogeneic Hematopoietic Stem Cell Transplant Recipients—A Retrospective Single-Center Analysis

**DOI:** 10.3390/vaccines11101534

**Published:** 2023-09-28

**Authors:** Alexander Nikoloudis, Ines Julia Neumann, Veronika Buxhofer-Ausch, Sigrid Machherndl-Spandl, Michaela Binder, Emine Kaynak, Robert Milanov, Stefanie Nocker, Olga Stiefel, Irene Strassl, Dagmar Wipplinger, Margarete Moyses, Heidrun Kerschner, Petra Apfalter, Michael Girschikofsky, Andreas Petzer, Ansgar Weltermann, Johannes Clausen

**Affiliations:** 1Department of Internal Medicine I: Hematology with Stem Cell Transplantation, Ordensklinikum Linz—Elisabethinen, Hemostaseology and Medical Oncology, 4020 Linz, Austria; 2Medical Faculty, Johannes Kepler University, 4020 Linz, Austria; 3Interdisciplinary Center for Infectious Medicine and Microbiology, Linz, Austria; 4Institute for Hygiene, Microbiology and Tropical Medicine, Ordensklinikum, Linz, Austria

**Keywords:** COVID-19, HSCT, vaccination, transplantation

## Abstract

(1) Background: mRNA COVID-19 vaccines are effective but show varied efficacy in immunocompromised patients, including allogeneic hematopoietic stem cell transplant (HSCT) recipients. (2) Methods: A retrospective study on 167 HSCT recipients assessed humoral response to two mRNA vaccine doses, using the manufacturer cut-off of ≥7.1 BAU/mL, and examined factors affecting non-response. (3) Results: Twenty-two percent of HSCT recipients failed humoral response. Non-responders received the first vaccine a median of 10.2 (2.5–88.9) months post-HSCT versus 35.3 (3.0–215.0) months for responders (*p* < 0.001). Higher CD19 (B cell) counts favored vaccination response (adjusted odds ratio (aOR) 3.3 per 100 B-cells/microliters, *p* < 0.001), while ongoing mycophenolate mofetil (MMF) immunosuppression hindered it (aOR 0.04, *p* < 0.001). By multivariable analysis, the time from transplant to first vaccine did not remain a significant risk factor. A total of 92% of non-responders received a third mRNA dose, achieving additional 77% seroconversion. Non-converters mostly received a fourth dose, with an additional 50% success. Overall, a cumulative seroconversion rate of 93% was achieved after up to four doses. (4) Conclusion: mRNA vaccines are promising for HSCT recipients as early as 3 months post-HSCT. A majority seroconverted after four doses. MMF usage and low B cell counts are risk factors for non-response.

## 1. Introduction

The onset of the coronavirus disease 2019 (COVID-19) pandemic, caused by severe acute respiratory syndrome coronavirus 2 (SARS-CoV-2), has undeniably put enormous stress on worldwide health systems. Individuals with compromised immunity, including patients with hemato-oncological diseases who have undergone allogeneic hematopoietic stem cell transplantation (HSCT), have been shown to be vulnerable to this virulent pathogen. Research has highlighted that the mortality rate for allogeneic HSCT patients hovers around 16%, as reported in a recent study [1]. This compelling statistic underscores the pressing urgency for developing and implementing effective preventative measures to safeguard this susceptible group.

The latter part of 2020 saw the advent of SARS-CoV-2 messenger ribonucleic acid (mRNA) vaccines, among which were the BNT162b2 and mRNA-1273 vaccines. The BNT162b2 and mRNA-1273 vaccines work by injecting a tiny fragment of the virus’s genetic code into the body. This triggers the production of the spike protein, which in turn activates the immune system. Once the immune system identifies the spike protein as an invader, it generates antibodies to combat it, thereby offering immunity against future infections with the virus [2]. Certainly, mRNA vaccines have revolutionized the field of immunization by offering superior effectiveness, minimal adverse reactions, and a streamlined, cost-efficient manufacturing process. These vaccines also hold the promise of transforming future medical treatments [3]. These were nothing short of a ray of hope in an otherwise dire global health scenario. The general population reported high efficacy from the mRNA-based vaccines [4,5,6], which sparked optimism. However, an area of uncertainty remained: the effectiveness of these vaccines in patients with hemato-oncological diseases, particularly in allogeneic HSCT recipients. This group’s immune systems are typically compromised due to the intense treatments they undergo, thereby posing potential challenges for the effectiveness of the vaccines. A recent study has shed some light on this issue, associating the vaccination with better outcomes in both autologous and allogeneic HSCT recipients [7].

Further analysis showed that allogeneic HSCT recipients developed anti-spike (anti-S) antibodies after receiving two or three doses of the vaccine [8]. This finding suggested that a significant proportion of these patients could potentially form an effective immune response to the vaccine. Despite this positive indication, other studies reported a lack of humoral responses in HSCT recipients [9]. This inconsistency accentuated the necessity to delve deeper into the various factors affecting the vaccine response in this highly vulnerable group.

Therefore, in the current study, our primary objective was to examine and assess the risk factors that could potentially contribute to a lack of humoral response in allogeneic HSCT patients post-vaccination. Specifically, we aimed to scrutinize these factors following the administration of two BNT162b2 mRNA or mRNA-1273 vaccine boosters within a cohort of 167 allogeneic HSCT recipients. The findings of this research could hold significant implications for the overall public health strategy concerning vaccination protocols in immunocompromised patients, particularly those who have undergone allogeneic HSCT.

## 2. Materials and Methods

This single-center retrospective research aimed to evaluate the rate of antibody response following a two-dose regimen of an mRNA vaccine for SARS-CoV-2. The study also sought to pinpoint the clinical and laboratory factors that might indicate a lack of effective antibody response in individuals who have undergone an allogeneic hematopoietic stem cell transplant (HSCT). The assessment of an effective antibody response was determined using a cut-off value set by the manufacturer at 7.1 binding antibody units (BAU)/mL or greater, as measured by the SARS-CoV-2 IgG II Quant assay from Abbott, Ireland [10,11].

The study population comprised all patients who had received an allogeneic blood stem cell transplant at the Ordensklinikum der Elisabethinen in Linz between 2 May 2003 and 12 March 2021 (163 of the 167 patients transplanted post-2011). The included participants were those who had undergone immunization (both first and second doses) by 12 October 2021 and had a minimum of 14 days between their second vaccination and titer measurement. 

In this study, the population was immunized using two types of COVID-19 vaccines: BNT162b2, which accounted for 88.0% (147 recipients), and mRNA-1273, which accounted for 12.0% (20 recipients). 

Several considerations were accounted for in the study. The last measured immunoglobulin (Ig) level and the most recently collected lymphocyte subtyping should not be older than 12 months for patients with a transplantation to first vaccination interval of up to 3 years. For patients with an interval of 3 to 5 years, the cut-off was increased to 18 months, and for those with an interval of >5 years the cut-off was 24 months. To account for potential confounding factors in recipients who recently received an immunoglobulin substitution, we ensured a minimum gap of 3 months since their last substitution.

The study examined various factors, such as the time elapsed between HSCT and vaccination, the interval from vaccination to antibody level measurement, the ages of both the recipient and the donor, ongoing cancer treatments, specific immunosuppressive medications, the kind of graft-versus-host disease (GVHD) prevention used, gender, the relationship between the donor and the recipient, the type of vaccine administered, and certain immune system indicators. These immune indicators included counts of different types of blood cells ((cluster of differentiation) CD4, CD8, and CD19 and (natural killer) NK cells) as well as levels of specific immunoglobulins (IgA, IgG, and IgM).

Our immunocompromised patient cohort was among the first in Austria to initiate SARS-CoV-2 vaccination. Therefore, passive transfer via transfusion is very unlikely to have contributed to the measured titers in our cohort, particularly to titers > 10^2^ or even 10^3^.

Statistical analyses included both univariate (response versus non-response) and multivariate logistic regression. Variables were excluded from the multivariate model until all variables were *p* < 0.2. Receiver operating characteristic (ROC) analysis was performed for CD19 cells with a cut-off estimation based on the Youden’s index. Comparative analysis of variables was conducted using the Fisher exact test and the Mann–Whitney U test. R statistical software was used for the analyses and plots used in this study [12].

## 3. Results

Table 1 details the characteristics of our cohort alongside the results from the univariate analyses, providing a comprehensive overview of the foundational data. Transitioning to a more visual interpretation, Figure 1 adeptly captures the outcomes of the multivariate model using a forest plot.

The study involved 167 adult (>18 years) participants who had received an allogeneic HSCT. The results showed that 37 of these individuals, making up 22% of the total study group, did not show a measurable antibody response after completing a two-dose mRNA vaccine series for SARS-CoV-2.

A detailed investigation into the vaccination non-responders revealed that the median time from HSCT to vaccination was 10.2 months, with a range of 2.5 to 88.9 months. In contrast, the median time interval for those who exhibited a response to vaccination (referred to henceforth as ‘responders’) was markedly longer at 35.3 months (range: 3.0 to 215.1 months). This difference in the time from HSCT to vaccination between non-responders and responders was statistically significant (*p* < 0.001). Given that the immune system requires time to recover post-transplant, these findings were anticipated. The multivariate model revealed an adjusted odds ratio (aOR) of 1.02 (confidence interval [CI] 95% [1.00–1.04], *p* = 0.12).

Age, which can significantly influence a patient’s response to treatment, demonstrated a difference between the two groups. The median age of the total cohort was 58.10 years, ranging from 23.40 to 76.60 years. The age difference was statistically significant with a *p*-value of 0.03 in the univariate analyses, suggesting that older age may be associated with a reduced response. In the multivariate analyses, this was only seen in a trend towards better response with younger age (aOR 0.96, CI 95% [0.22–1.00], *p* = 0.06).

The study also observed differences in gender distribution, although these differences were not statistically significant with a *p*-value of 0.06. When subjected to multivariate analyses, no significant disparities were evident (aOR 0.44, CI 95% [0.14–1.35], *p* = 0.15). It should be noted that, in our study, a slightly higher number of males than females were evaluated. 

By univariate analysis, we observed significant differences in lymphocyte subset counts between responders and non-responders. For CD4+ cells, the median count was 386 cells/μL (ranging from 79 to 1629) in responders, compared to 154 cells/μL (ranging from 7 to 1007) in non-responders, with a highly significant *p*-value of <0.001. Similarly, CD8+ cells showed a median count of 488 cells/μL (33 to 4654) in responders and 295 cells/μL (44 to 2473) in non-responders, with a *p*-value of 0.01. For CD19+ cells, the median count was 295 cells/μL (0 to 1295) in responders, significantly higher than the 17 cells/μL (0 to 530) in non-responders, with a *p*-value of <0.001. NK cells showed a minimal difference, with median counts of 250 cells/μL (15 to 1979) in responders and 218 cells/μL (14 to 646) in non-responders, with a *p*-value of 0.05 (Table 1). These results indicate stronger cellular immunity in the responder group for CD4+, CD8+, and CD19+ cells.

However, only the CD19+ B cell count remained a significant variable in the multivariate model. Here, the likelihood of a humoral response to vaccination was found to be associated with a higher CD19-positive cell count. For every increase of 100 CD19-positive cell/μL, the aOR for a humoral vaccination response was 3.3 (CI 95% [1.78–6.20], *p* < 0.001). This observation underscores the pivotal role of B-cells in mediating a robust humoral response to vaccination. Upon conducting an ROC analysis, a threshold of 146.5 cells/μL yielded an area under the curve (AUC) of 89.3% (CI 95% [83.4–95.2%]; Figure 2).

Additionally, by univariate analysis, we observed notable differences in immunoglobulin levels between responders and non-responders. For IgA, the median level was 92.00 mg/dL (ranging from 5.00 to 454.00) in responders, compared to 51.00 mg/dL (ranging from 1.00 to 1299.00) in non-responders, with a significant *p*-value of 0.001. IgG levels also showed a marked difference, with a median level of 828.50 mg/dL (72.00 to 2137.00) in responders and 436.00 mg/dL (161.00 to 2079.00) in non-responders (*p* < 0.001). Similarly, IgM levels were higher in responders, with a median level of 62.50 mg/dL (5.00 to 299.00), as opposed to 33.00 mg/dL (1.00 to 142.00) in non-responders, also with a highly significant *p*-value of <0.001. While these differences were statistically significant in the univariate analyses, they did not remain significant in the multivariate model. 

Out of the 37 non-responders, 34 (92%) received a third dose of one of the two available mRNA-based vaccines. This was carried out at a median of 189 days after the second vaccination (range: 56 to 273 days). One patient declined revaccination, while two other patients were not eligible for revaccination due to the progression of malignancy, which unfortunately proved to be fatal (Figure 3).

Of the 34 individuals who did not respond to the initial two doses and went on to receive a third dose of the vaccine, 23 out of the 30 who were evaluated (77%) showed a successful antibody response, i.e., achieved seroconversion. For the seven patients who did not achieve seroconversion even after the third dose, six were administered a fourth mRNA vaccine dose. This was given at an average of 95 days after the third dose, and it resulted in a 50% additional seroconversion rate.

Among the 23 patients who did achieve seroconversion following the third dose, 10 (44%) went on to receive a fourth mRNA vaccine. This was administered a median of 125 days after the third dose. The median antibody level for these 23 patients, as measured by the SARS-CoV-2 IgG titer, clearly increased from 290.8 BAU/mL after the third dose to 1750 BAU/mL after the fourth dose.

By the end of the observation period, a total of 25 out of the initial 37 patients who failed the basic two-dose immunization were able to produce detectable levels of SARS-CoV-2 IgG antibodies. Thus, we demonstrated that the seroconversion rate of 78% after a two-fold immunization could be increased to a cumulative rate of 93% by two additional vaccines. This finding underscores the potential benefit of additional vaccine doses in inducing a detectable antibody response among HSCT recipients.

In our study cohort, we observed a total of 39 documented cases of COVID-19 infection. Among these, 11 cases were found in the non-responder group, while the remaining 28 cases were identified among patients with documented seroconversion. In the cohort of non-responders, the infection rate was observed to be 29.7%, whereas in the responder group, the rate was notably lower at 21.8%. Statistical analysis revealed that the difference in infection rates between the non-responder and responder groups was not significant, with a *p*-value of 0.38. It is noteworthy that only 2 out of the 39 patients with confirmed COVID-19 infection required hospital admission. Specifically, one patient was from the non-responder group and the other was from the responder group. No deaths related to COVID-19 infections were observed in the study population. Admission to the intensive care unit was not needed for any of the patients.

The distribution of immunosuppressive regimens showed significant differences. While 60% of responders were off immunosuppression, a mere 18.9% of non-responders were off these drugs. Regimens like MMF combined with or without CNI were more prevalent in non-responders (Table 1). Specifically, our analysis indicated that concurrent immunosuppression therapy with mycophenolate mofetil (MMF) reduced the likelihood of a response to vaccination. The aOR of responding to the vaccine was as low as 0.04 (CI 95% [0.01–0.24], *p* < 0.001) in patients undergoing MMF therapy at the time of vaccine. On the other hand, calcineurin inhibitors (cyclosporine A or tacrolimus) as standalone immunosuppressive therapy did not show an impact on the outcome, with an aOR of 1.15 (CI 95% [0.22–5.98], *p* = 0.87).

Further, it was observed that ruxolitinib when administered alone did not show a significant influence on the vaccination success, with an aOR of 0.25 (CI 95% [0.04–1.67], *p* = 0.15). However, there was a trend in the multivariable analysis when ruxolitinib was combined with calcineurin inhibitors, hinting towards a negative impact on vaccination outcomes with this combination, with an aOR of 0.12 (CI 95% [0.01–1.00], *p* = 0.05).

Inclusion of ATG for primary GVHD prophylaxis was a significant risk for non-response in the multivariate model, with an aOR of 0.26 (CI 95% [0.07–0.94], *p* = 0.04).

Unrelated donor transplantations were slightly more prevalent among non-responders (40.5%) than responders (30.8%). However, this difference was not statistically significant (*p* = 0.32; Table 1).

In the univariate analyses, we found no significant differences between the mRNA-1273 and BNT162b2 vaccines when comparing the responder group to the non-responder group after two doses. Specifically, among the 167 participants, 147 (88.0%) received the BNT162b2 vaccine and 20 (12.0%) received the mRNA-1273 vaccine. In the responder group (n = 130), 115 (88.5%) received BNT162b2 and 15 (11.5%) received mRNA-1273. In the non-responder group (n = 37), 32 (86.5%) received BNT162b2 and 5 (13.5%) received mRNA-1273. The *p*-value for the difference between the vaccines in terms of response was 0.78, indicating no significant difference.

A significant majority of responders (76.9%) were not on steroids, compared to only 35.1% of non-responders. This indicates that ongoing steroid treatment might negatively affect the response, a finding that was statistically supported (*p* < 0.001). This could not be supported by the multivariate model, where steroid therapy was not selected in the final model (Figure 1). Ongoing maintenance treatment of the underlying malignancy was well-balanced among both groups, with around 16% in each group undergoing maintenance treatment at the time of the first vaccine. 

## 4. Discussion

This retrospective single-center study, conducted to assess the humoral response rate to a two-dose primary mRNA vaccination against SARS-CoV-2 in recipients of an allogeneic HSCT, provides valuable insights into the factors influencing the immune response for recipients of allogeneic hematopoietic stem cell transplantation. The study found that 22% of the study population failed to mount a detectable antibody response following two-fold vaccination. The likelihood of a humoral response was found to be associated with a higher CD19-positive cell count, highlighting the crucial role of B-cells in ensuring an effective humoral response to the vaccination. Concurrently, the study indicated that concurrent immunosuppression therapy reduced the likelihood of a response to vaccination.

These findings are consistent with previous research that has shown that the immune response to SARS-CoV-2 vaccination after HSCT or solid organ transplantation can be influenced by a variety of factors, including the type of vaccine, the timing of vaccination relative to transplantation, and the use of immunosuppressive therapy [13,14,15,16,17,18,19]. Also, the role of B-cells in the immune response to vaccination has been described before [14,20,21]. In accordance with our findings, one large prospective study revealed the B cell count as predictive for humoral response after the second vaccine, but not after a third or fourth vaccine dose, and additionally demonstrated the feasibility of initiating vaccination during the first 4 months after allogeneic HSCT [22].

Our study also found that, by univariable analysis, the median time from HSCT to vaccination was significantly shorter in non-responders compared to responders. This is in line with previous studies that have suggested that the immune system may need time to recover following HSCT before it can respond effectively to vaccination [15,23]. However, since the time from HSCT did not remain a significant risk factor for non-response in the multivariable analysis, while a low B-cell count and particular immunosuppressives such as MMF did, a shorter time from HSCT per se should not be considered a contraindication to beginning a vaccination series. Indeed, the first vaccination in the responder cohort was as early as 3 months after HSCT.

The association between immunosuppression and an insufficient humoral response to COVID-19 vaccination has been described before. In our analysis, MMF was the strongest negative predictor among the analyzed regimens in the multivariate model. This negative association of MMF with humoral response has also been seen in other studies on HSCT and kidney transplantation [18,24] This finding highlights the importance of carefully managing immunosuppressive therapy in HSCT recipients receiving SARS-CoV-2 vaccinations and may also indicate the potential benefit of an additional vaccine dose for this population.

In our study, it was observed that patients treated with ruxolitinib alone did not exhibit a significantly higher rate of non-responsiveness to the vaccine. However, when combined with calcineurin inhibitors, there was a noticeable trend. One source indicates that ruxolitinib does not impair the humoral immune response to the BNT162b2 mRNA COVID-19 vaccine in patients with myelofibrosis [25]. Another study that investigated ruxolitinib use in patients with myelofibrosis discovered that it was associated with suboptimal response to the vaccination [26]. It is noteworthy to mention that both of these studies showed different indications to GVHD in patients after allogeneic HSCT, as it was used in our study.

Interestingly, our findings reveal that a significant proportion of non-responders were able to produce detectable levels of SARS-CoV-2 IgG antibodies following additional doses of the vaccine. This suggests that additional doses of the vaccine may be beneficial in inducing a detectable antibody response in HSCT recipients who fail to respond to the initial two-dose regimen. This is in line with recent observations of the antibody response to the third vaccination [27,28,29].

In a previous study, there was evidence suggesting that the female gender had a more favorable response to the SARS vaccination, especially following the third dose, in recipients of allogeneic HSCT [29]. However, in our comprehensive analysis, both at the univariate and multivariate levels, this purported gender advantage did not manifest any statistically significant differences. Further research is essential to elucidate the intricacies of the relationship between gender and vaccine efficacy in this specific cohort. 

The role of CD19+ cells in predicting outcomes has been a topic of interest in numerous clinical trials [14,20,21]. Our research, when contrasted with the findings from Schulz et al. [21], presents some key differences. Schulz et al. [21] reported CD19+ naïve B cells as a predictor for humoral response, particularly in immunocompromised patients. Interestingly, our ROC analysis unveiled a cut-off that is approximately 2.5-fold higher than theirs. A plausible explanation for this discrepancy could be the distinct nature of the cohorts. While Schulz et al. encompassed a varied patient demographic, our study specifically centered on allogeneic HSCT recipients. 

Another noteworthy differentiation between the two studies is the choice of the vaccine administered. A significant portion of our participants received the BNT162b2 vaccine. In contrast, the cohort in Schulz et al. [21] was mainly administered the mRNA-1273 vaccine. It is essential to acknowledge that, in our study, there was a wider range of CD19 measurements. Unlike Schulz et al. [21], we did not delve deeper into classifications beyond the general marker of CD19 positivity. The absence of this detailed stratification could potentially have influenced the divergent outcomes between the two research studies.

Further adding to this discussion, another study scrutinizing allogeneic HSCT patients highlighted various factors influencing vaccine inefficacy post-vaccination for different clinical conditions. Remarkably, the authors’ multivariate analysis indicated B-cell counts below 135 per microliters (mcl) as a pivotal predictor for vaccine failure a year after transplantation [30]. This finding aligns closely with our ROC analysis cut-off, reinforcing the importance of CD19+ cell counts in vaccine efficacy evaluations.

In light of the above, it is essential to discuss the key factors that might explain the differences between our findings and those presented by Schulz et al. [21]. Primarily, the cohorts in the two studies had different characteristics. Schulz et al.’s study [21] featured a more diverse patient demographic, whereas ours was primarily concentrated on allogeneic HSCT recipients. Additionally, the vaccines used in the two studies were different, which could have inherently influenced the outcomes. Finally, the absence of a more granular analysis of CD19+ cells in our study, as compared to Schulz et al. [21], who further classified these cells, could have been a contributory factor to the discrepancies observed in the comparative findings between the two studies. 

No vaccine-related complications [2,31] were recorded after the date of the first vaccinations, e.g., thromboembolic events or myocarditis, in all recipients. 

Despite the valuable insights provided by this study, several limitations should be acknowledged. The retrospective design may have introduced bias, and the absence of a control group limits the ability to draw definitive conclusions. At the time of the described vaccination program, COVID-19 infections were still very rare in our highly isolated and protected vulnerable patient cohort. Since we also had documented individual infections in the cohort, only five patients had a documented infection before completing the described vaccination series. The wide range in the timing of vaccination relative to transplantation and titer measurement could have introduced confounding factors. The lack of long-term follow-up data restricts the assessment of the durability of the antibody response and the vaccine’s clinical effectiveness. While this study provides valuable insights into the humoral response to vaccination, it is important to note that it does not fully capture the complexity of the immune response to SARS-CoV-2, particularly the role of T-cell responses. Indeed, recent research underscores the robustness of T-cell responses to vaccination, even in the absence of detectable antibodies [32,33]. Lastly, the inclusion of different types of mRNA vaccines, specifically BNT162b2 and mRNA-1273, could have introduced variability in the results. We chose to limit our study to mRNA vaccines due to their widespread availability and comparable mechanisms of action, which allowed for a more focused analysis.

## 5. Conclusions

In light of our comprehensive retrospective single-center analysis, the factors influencing the immune response to SARS-CoV-2 mRNA vaccination in recipients of allogeneic HSCT extend beyond simple post-transplantation timelines. A myriad of aspects, including concurrent immunosuppression therapies, particularly the use of MMF, CD19-positive B-cell counts, and even the choice of vaccine administered, play pivotal roles in determining humoral response efficacy. Notably, while a significant section of non-responders demonstrated a marked response to additional vaccine doses, the complexity of the immune response to SARS-CoV-2 cannot be captured by antibody response alone. T-cell responses, as recent research suggests, offer a robust reaction to vaccination even without detectable antibodies. This underlines the need for a holistic, multifaceted approach in tailoring vaccination strategies for allogeneic HSCT recipients. Careful consideration of immunosuppressant regimens, timely evaluations of B-cell counts, and perhaps even the nature of the vaccine itself may be crucial determinants. Nonetheless, while our findings provide significant insights and are supported by comparative studies, the retrospective nature and potential confounding factors necessitate cautious interpretation. Prospective, controlled studies will further elucidate and solidify the directions indicated by our research.

## Figures and Tables

**Figure 1 vaccines-11-01534-f001:**
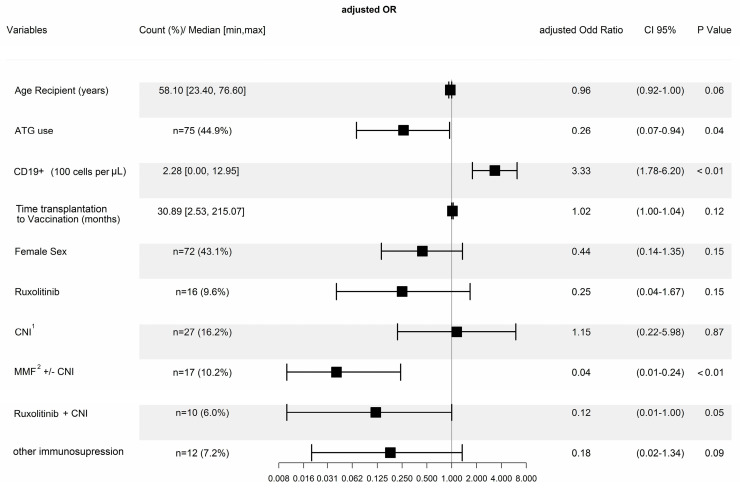
Forest plot showing the results of the multivariate logistic regression model (humoral responders versus non-responders). ^1^ Calcineurin inhibitors; ^2^ mycophenolate mofetil.

**Figure 2 vaccines-11-01534-f002:**
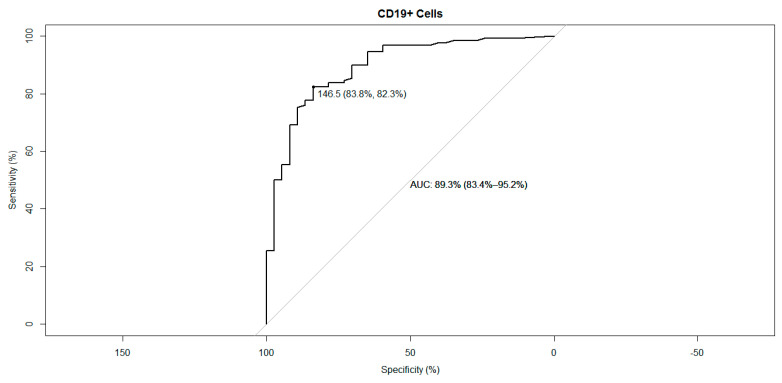
The receiver operating characteristic (ROC) analysis curve for CD19-positive cells between humoral responders and non-responders after two mRNA SARS-CoV-2 vaccinations.

**Figure 3 vaccines-11-01534-f003:**
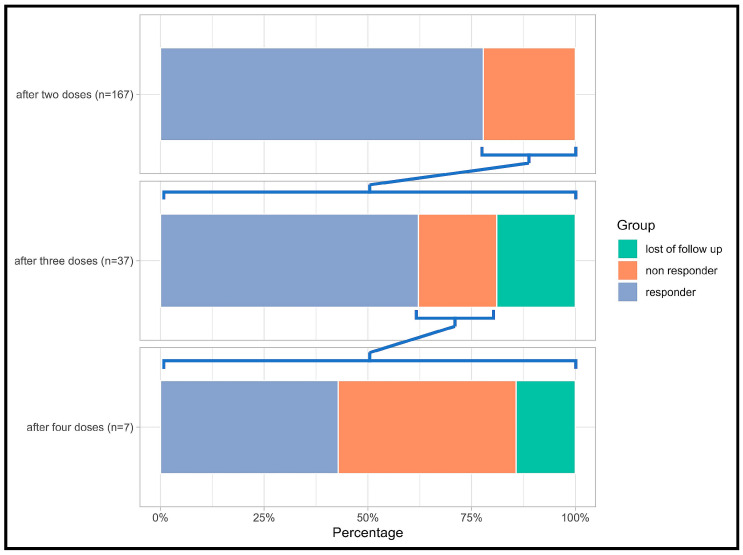
Illustration of humoral responses to mRNA vaccinations. The chart categorizes individuals as responders (purple), non-responders (orange), and those lost to follow-up (green; e.g., those declining further vaccinations). The data are segmented into three bar charts representing the number of vaccination doses received. Notably, after each dose, only non-responders are charted for subsequent doses to evaluate their responsiveness.

**Table 1 vaccines-11-01534-t001:** Cohort characteristics and univariate analyses of humoral responders and non-responders after the second mRNA vaccination.

		All	Responders	Non-Responders	*p*-Value
		(n = 167)	(n = 130)	(n = 37)	
Baseline characteristics	Unit				
Age (median [min, max])	Years	58.10 [23.40, 76.60]	57.20 [23.40, 76.50]	64.30 [23.50, 76.60]	0.03
Sex (%)					
	Male	95 (56.9)	79 (60.8)	16 (43.2)	0.06
	Female	72 (43.1)	51 (39.2)	21 (56.8)	
Vaccine (%)					
	BNT162b2	147 (88.0)	32 (86.5)	115 (88.5)	0.78
	mRNA-1273	20 (12.0)	5 (13.5)	15 (11.5)	
Time from transplantation to vaccination (median [min, max])	Months	30.89 [2.53, 215.07]	35.26 [2.99, 215.07]	10.20 [2.53, 88.88]	<0.001
Unrelated donor (%)					
	Related	112 (67.1)	90 (69.2)	22 (59.5)	0.32
	Unrelated	55 (32.9)	40 (30.8)	15 (40.5)	
Antineoplastic maintenance treatment (%)					
	No	141 (84.4)	110 (84.6)	31 (83.8)	1.00
	Yes	26 (15.6)	20 (15.4)	6 (16.2)	
Ongoing steroid treatment (%)					
	No	113 (67.7)	100 (76.9)	13 (35.1)	<0.001
	Yes	54 (32.3)	30 (23.1)	24 (64.9)	
Immunosuppressives (%)					
	Off immunosuppression	85 (50.9)	78 (60.0)	7 (18.9)	<0.001
	Ruxolitinib	16 (9.6)	12 (9.2)	4 (10.8)	
	CNI ^1^ (CSA ^2^ or TAC ^3^)	27 (16.2)	22 (16.9)	5 (13.5)	
	MMF ^4^ +/− CNI	17 (10.2)	4 (3.1)	13 (35.1)	
	Ruxolitinib + CNI	10 (6.0)	7 (5.4)	3 (8.1)	
	Other IST ^5^	12 (7.2)	7 (5.4)	5 (13.5)	
Lymphocyte counts (median [min, max])	(Cells/μL)				
	CD4+	354 [7, 1629]	386 [79, 1629]	154 [7, 1007]	<0.001
	CD8+	459 [33, 4654]	488 [33, 4654]	295 [44, 2473]	0.01
	CD19+	228 [0, 1295]	295 [0, 1295]	17 [0, 530]	<0.001
	NK	239 [14, 1979]	250 [15, 1979]	218 [14, 646]	0.05
Immunoglobulin levels (median [min, max])	(mg/dL)				
	IgA	79.00 [1.00, 1299.00]	92.00 [5.00, 454.00]	51.00 [1.00, 1299.00]	0.001
	IgG	760.00 [72.00, 2137.00]	828.50 [72.00, 2137.00]	436.00 [161.00, 2079.00]	<0.001
	IgM	55.00 [1.00, 299.00]	62.50 [5.00, 299.00]	33.00 [1.00, 142.00]	<0.001

^1^ Calcineurin inhibitors; ^2^ cyclosporine *A*; ^3^ tacrolimus; ^4^ mycophenylate mofetil; ^5^ immunosuppressive therapy. NK, natural killer.

## Data Availability

The data presented in this study are available upon request from the corresponding authors. The data are not publicly available due to restrictions concerning privacy or ethical considerations.

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
