# Peer review of "Successful SARS-CoV-2 mRNA Vaccination Program in Allogeneic Hematopoietic Stem Cell Transplant Recipients—A Retrospective Single-Center Analysis"

_vaccines, 2023, doi:10.3390/vaccines11101534_

Round 1

Reviewer 1 Report

The present work by Nikoloudis et al. shows the results of a retrospective study on SARS-CoV-2 mRNA vaccination outcomes in patients who underwent allogeneic Hematopoietic Stem Cell Transplant: the authors evaluated anagraphical, laboratory and clinical data, that could have influenced immunization to help in planning vaccination schedule in this population and maximize seroconversion.

The work is clear and understandable, with no need of improvement in its form.

Here follows some specific consideration.

- Review the use of acronyms: sometimes the extended form is not present or placed the first use of acronym

- Modify table 1: columns are not not in line with titles

- Review the Discussion section: there are repetitions (lines 270-277 and 283-290), that make difficult to understand the text; in addition, confounding sentences are present in the section (lines 274-276)

Author Response

Sincerely yours,

Dr. Nikoloudis and colleagues

Reviewer 2 Report

In my review of the manuscript “Successful SARS-CoV-2 mRNA Vaccination Program in Allogeneic Hematopoietic Stem Cell Transplant Recipients – a Retrospective Single-Center Analysis” the author needs to correct the spelling and sentence forming errors, and enrich each section with including more details about the study, in this form this manuscript not acceptable for publication, follow the below comments and solve them, it will help to enrich the content of this manuscript.

Minor comments:

1.      Add the full form of COVID-19.

2.      Add the reference when use quantitative data.

3.      Add reference at line 72.

4.      Author should make sure that, add sufficient information in sentence formation.

Major comments:

1.      In the result section, add the detailed result of each study.

2.      Conclusion not up to the mark, modify it.

3.      Author should refer below articles for further modifications, and add details from this.

https://doi.org/10.1186/s13037-021-00291-9 and https://doi.org/10.3390/vaccines10122150

Author Response

(The authors gave the same response as above.)

Reviewer 3 Report

I have only minor comments:

L78: What were the vaccines used for the immunization of this study population? Please provide the details on vaccine characteristics in this section. There are details in Table 1 but a solid sentence is missed in the M&M section.

L100: The main concern I have regarding these results is related to the missing transfusion rate. It is common knowledge that allogeneic HSCT recipients require several (or numerous) blood transfusions during the post-HSCT period. Is it possible that SARS-CoV-2 specific antibodies could be of external (blood transfusion-based) origin, considering the pandemic setting of this study? Also, do you have any data on the serostatus of allo-HSCT donors for these patients?

L105: Please add 'adult (>18 years)' between the words '167' and 'allogeneic'.

L112-113: This finding makes sense, since the immune response recovery requires time.

L136: Please delete the 'the model of'.

L197 (Table 1): Re-formatting is needed for the Table 1, Responders and Non-responders as well as P-value columns need to be separated accordingly.

L230-232: This sentence needs to be rephrased in order to improve clarity.

L246-248: Or perhaps to introduce an additional vaccine dose?

L263-268: In addition, females were a slight minority in this study population.

L270: Please change 'predictive' to 'protective'.

L294-295: The absence of the control group is perhaps the biggest limitation of this study.
Also, what about the BTI (breakthrough infection) data? Is it possible that, in the pandemic setting, detected anti-S Ab are due to infection in vaccinated allo-HSCT recipients (or perhaps, blood transfusion)?

L303-304: What kind of different types? And why limiting only to the mRNA vaccines?

Author Response

(The authors gave the same response as above.)

Round 2

Reviewer 2 Report

The authors have addressed all my suggestions. Now the manuscript looks in better shape.